
# Formation characteristics of aerosol triplet state and coupling effect between the separated components with different polarity

Dongjie Guan[a], Qingcai Chen[a*], Jinwen Li[a], Hao Li[a], Lixin Zhang[a], Yuqin Wang[a], Xiaofei Li[a] and Tian Chang[a]

*[a] School of Environmental Science and Engineering, Shaanxi University of Science and Technology, Xi'an 710021, China*

*Corresponding author:

School of Environmental Science and Engineering, Shaanxi University of Science and

Technology, Weiyang District, Xi'an, Shaanxi, 710021, China

*(Q. C.) Phone: (+86) 0029-86132765; e-mail: chenqingcai@sust.edu.cn;





**Abstract:** Atmospheric aerosols contain organic matter that can form triplet state ($^3$C*)
excited by sunlight, which plays a critical role in the aging process of aerosols. In order to
understand the triplet state reaction mechanism of complex aerosol components, the
formation characteristics of $^3$C* in the aerosol components with different polarity, i.e., the
highly polar water-soluble matter (HP-WSM), humic-like substances (HULIS) and
methanol-soluble matter (MSM) were investigated. The coupling effect of generation of $^3$C*
and reactive oxygen species (ROS) between different aerosol components was also
examined. The results show that the $^3$C* generation characteristics is strongly dependent on
the polarity of these components. HULIS has the strongest generation ability of $^3$C*, and the
MSM contribute the most to the total generation of $^3$C*. It is found that the high-energy
triplet states ($E_T \geq 250$ kJ mol$^{-1}$) of HULIS and HP-WSM components account for up to 80%.
These $^3$C* has an important contribution to the photochemically generation of ROS, and the
generated ROS of different components are also different, which is determined by the
chromophore composition of complex organic matter. Tyrosine-like chromophore is the
main substance leading to the formation of water-soluble $^3$C*, whilethe highly oxidized
HULIS chromophore plays a leading role in the water-insoluble component. This study alos
found that there is a coupling effect between HP-WSM and HULIS on $^3$C* and ROS
generation. The $^3$C* generation rate increases by about 40% after mixing, but the generation
of $^1$O$_2$ is severely reduced. Overall, this study provides deep insights into the generation
characteristics of the triplet state of atmospheric aerosols.
**Key words:** atmospheric aerosols; photochemistry; triplet state; reactive oxygen species;
coupling effect
**TOC:**

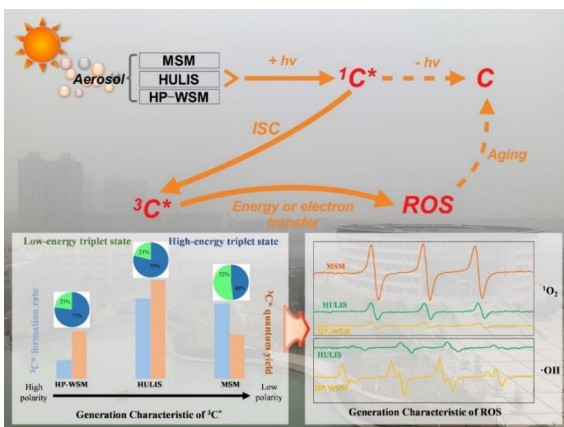



## 1. Introduction

Atmospheric photochemical processes play an important role in the formation and
aging of aerosols (Atkinson, 2000; Derwent et al., 1998; Lim et al., 2005; Ervens et al., 2011;
Blando and Turpin, 2000). For example, the photo-oxidation process of organic matter and
nitrogen oxides affect the cycle of carbon and nitrogen elements (Rollins et al., 2012;
Goldstein and Galbally, 2007), and the volatile organic compounds undergo photochemical
reactions to generate secondary organic aerosols (SOA) (Griffith et al., 2013; Arce et al.,
2008). Part of the photochemical reaction of aerosol is direct photolysis, and in most cases, it
is driven by photochemically generated reactive intermediates, including triplet organics
($^3C^*$) and reactive oxygen species (ROS) (Warneck, 1999; Herrmann et al., 2015; Rincon et
al., 2009; Lee et al., 2014).
Environmental $^3C^*$ is discovered earlier in natural water, and it has been proven to be
an important environmental photochemical reaction intermediate (Vione et al., 2014; Xu et
al., 2011). After absorbing solar radiation, this kind of substance will transition from the
ground state to the excited singlet state ($^1C^*$), and then rapidly transition to $^3C^*$ through the
intersystem. $^3C^*$ directly participates in the degradation of pollutants through the process of
energy transfer or electron transfer (Canonica et al., 2006; Parker et al., 2013). It can also
indirectly degrade pollutants by generating ROS ($^1O_2$, $\cdot OH$, $O_2^{\cdot-}$, $\cdot HO_2$, etc.) (Wenk et al.,
2011; Garg et al., 2011; Glover et al., 2013). Because the lifetime of $^1C^*$ is much shorter
than that of $^3C^*$, the concentration of $^3C^*$ in the environment is higher than that of $^1C^*$, and
the impact of $^3C^*$ on the environment is also greater (McNeill and Canonica, 2016).
Compared with the water environment, there are fewer studies on the triplet state in the
atmospheric environment. Kaur and Anastasio (2018) used probe technology to conduct
photochemical experiments on atmospheric fog droplets and confirmed the existence of $^3C^*$
in the atmospheric environment for the first time. Atmospheric chromophores are organic
substances with light-absorbing properties in the atmospheric environment, which are the
precursors of $^3C^*$ in aerosols. For example, imidazoles and pyrazines that are widely present
in the atmosphere can absorb light radiation to form $^3C^*$ (De-Haan et al., 2010; Hawkins et
al., 2018; Laskin et al., 2015). However, because the atmospheric chromophore is not a





single substance, but a complex organic mixture, this also increases the difficulty of
studying the generation mechanism of $^3C^*$ in actual atmospheric environment.
The environmental organic matter components are composed of complex compounds
with different polarities, which are expected to have different optical properties and
photochemical reactivity. A study explored the differences in the optical properties of
aerosol components with different polarities, and found that substances with lower polarities
have a stronger ability to absorb light radiation (Chen et al., 2017). Zhou et al. (2017)
separated three organic components of hydrophobic, hydrophilic and transitional hydrophilic
(medium hydrophilic) from polluted wastewater, and found a significant difference between
$^3C^*$ quantum yield coefficient ($f_{TMP}$) and singlet oxygen quantum yield ($\Phi_{1O2}$) of each
component. $^3C^*$ converts $O_2$ into $^1O_2$ by means of energy transfer, so this process can be
used to determine the energy ($E_T$) distribution of $^3C^*$, which can be divided into low-energy
triplet states (94 kJ mol$^{-1}$ ≤ $E_T$ < 250 kJ mol$^{-1}$) and high-energy triplet states ($E_T$ ≥ 250 kJ
mol$^{-1}$) (Wilkinson et al., 1993; Kellogg and Simpson, 1965). Zhou et al. (2019) found that
high-energy $^3C^*$ accounted for more of the organic matter in water, while soil organic matter
had more low-energy $^3C^*$. Although there are some researches on the characteristics of $^3C^*$
formation in water environments, there are very few studies on the characteristics of $^3C^*$
types in the atmospheric environment. Due to the large differences in the sources and
chemical processes of organic matter in the atmospheric environment and the water
environment, the research conclusions on the $^3C^*$ generation characteristics of water bodies
may not be suitable for atmospheric aerosols.
Three-dimensional matrix excitation-emission (EEM) fluorescence spectroscopy
technology has been widely used to explore the composition of complex atmospheric
chromogenic organic matter (Chow et al., 2004). Chen et al. (2020) used EEM technology to
establish the relationship between the atmospheric chromophores and the types and sources
of substances, and expanded the application of EEM in the field of atmosphere. The cause of
the formation of $^3C^*$ is directly related to the chromophore. Therefore, the EEM method
may be conductive to revealing the chemical mechanism of aerosol triplet states. ROS is an
important driving factor in the aerosol aging process. Many studies have shown that $^3C^*$ can
drive the generation of ROS, and the generation characteristics and mechanisms of ROS by





different $^3C^*$ species are not yet known (Vione et al., 2006; Perri et al., 2009;). In summary,
in order to explore the formation characteristics and mechanism of triplet in atmospheric
aerosols, this study separated complex aerosol components according to polarity, and
obtained highly polar water soluble (HP-WSM), humic-like substances (HULIS) and
methanol soluble components (MSM, representative of water-insoluble organic matter). Use
chemical probe methods (including 2,4,6-trimethylphenol, furfuryl alcohol, and sorbic
alcohol) to characterize the generation characteristics of triplet states of different
components, including $^3C^*$ generation rate, quantum yield coefficient, and energy
distribution (Zhou et al., 2017; Zhou et al., 2019). This study also explored the
structure-activity relationship between the type of chromophore and the generation of $^3C^*$,
and demonstrated that the generation characteristics and mechanism of ROS depends on the
generation of $^3C^*$. Finally, this study confirmed the coupling effect of generation of $^3C^*$ and
ROS between different aerosol components (HP-WSM and HULIS).
**2. Experimental Section**
*2.1 Experimental materials*

A $Mn^{2+}$ standard in ZnS and $Cr^{3+}$ standard in MgO were purchased from Freiberg

Instruments        Inc.,        Delfter,        Germany.        Glucose        (≥99.5%),
4-Hydroxy-2,2,6,6-tetramethylpiperidine (TEMP, ≥98%), L-Histidine (≥99%), Phenol
(≥99.5%), 2,4,6-Trimethylphenol (TMP, ≥98%), Furfuryl alcohol (FFA, 98%) and
5,5-Dimethyl-1-pyrroline-N-oxide (DMPO, 97%) were purchased from Aladdin Reagent
Company (Shanghai, China). Trans, trans-2,4-hexadien-1-ol (Sorbic alcohol, 97%) was
purchased from Sigma-Aldrich. 4-Nitroanisole (PNA, ≥99%) was purchased from
Thermo-Fisher. Pyridine (pyr, ≥99%) was purchased from Alfa-Aesar. C18 cartridge (500
mg/6 mL) was obtained from Agela Technologies. All chemicals were used as received.
*2.2 Sample collection and preparation*

Atmospheric $PM_{2.5}$ samples were collected in Xi'an city, China during the winter from

2019.12.21 to 2020.2.1 (Sample list is shown in **Table** S1). The sample collection point was
located on the top of Shaanxi University of Science and Technology, about 40 m above the
ground. A large flow sampler (XT-1025, Shanghai Xintuo, China) was used to collect



atmospheric particulate samples for 23.5 hours. The samples were collected on a 20×25 cm

quartz membrane pre-fired in a muffle furnace (2500 QAT-UP, Pallflex Products Co., US),

then the collected sample film is stored at -20 ℃ until use. The simulated combustion PM

samples include wheat straw, rice straw, poplar wood, bituminous coal, and lignitous coal.

The combustion samples were prepared through a self-built combustion-gas-gathering

device. The final particles were collected on a 47 mm diameter quartz membrane and stored

at -20 ℃ (**Text S1** for specific sample collection process and sample information).

The samples were continuously extracted with water and methanol ultrasonically for 20

minutes, and then filtered with a 0.45 µm PTFE filter to obtain the water-soluble extract

(WSM) and methanol extract (MSM), respectively. Note that the MSM here does not

actually contain water-soluble substances, thus it represents water-insoluble organic matter.

Then, according to the previous method,the solid-phase extraction technology was used

extract HP-WSM and HULIS from WSM (Lin et al., 2012; Chen et al., 2016), as shown in

**Text** S1 (the separation process) and **Table** S3 (sample component information). The final

concentration of all solution samples after dilution or concentration is controlled to 20 mg-C

$L^{-1}$, and stored at 4 ℃ for later use.

*2.3 Organic carbon (OC)/elemental carbon (EC) analysis*

Both original sample films and the prepared sample solution were used for the analysis

of organic carbon (OC)/elemental carbon (EC), and an OC/EC analyzer (Model 4, Sunset,

America) with IMPROVE_A temperature protocol was employed for quantitative analysis.

The original sample was directly measured with a 6 mm diameter filter membrane, and the

sample solution was subjected to OC/EC analysis by loading 100 µL of the solution onto the

17 mm diameter membrane, and then drying the solvent with nitrogen blowing. The

background filter sample was also processed and analyzed with the same procedure, and the

background signal interference was subtracted from the final result.

*2.4 Spectroscopic characterization*

The sample's absorption spectrum and fluorescence spectrum were obtained at the same

time in the "fluorescence + absorbance" mode by an Aqualog EEM analyzer (Horiba

Scientific, USA). In order to reduce the influence of internal filtration effect on EEM

measurement, the solution sample used for optical analysis was diluted before analysis. The





concentration of the solution during analysis is shown in **Table** S4. The instrument analysis
parameters are set as follows: excitation wavelength of 200-600 nm, emission wavelength of
250-800 nm, scanning interval of 2 nm and exposure time of 1 s. The background sample
was also analyzed in the same way and subtracted from the actual sample spectrum. A total
of 80 sample EEM spectra were analyzed through the PARAFAC model to identify the
chromophore types. According to the change trend of the minimum residual error of the 2-7
component PARAFAC model (**Fig.**S10) and the interpretability of the chromophores to the
actual meaning, finally 4 component model was used (**Fig.**S11).
*2.5 Photochemical experiments*
The sunlight is simulated by a 300 W xenon light source (PLS-SXE 300, Perfectlight,
China) and equipped with a VISREF filter. By combining a cold water circulation machine
and a magnetic stirrer, the experiment temperature is kept at about 25 °C. The sample
solution (120 μL) added with the chemical probe is placed in a special quartz reaction dish
(diameter 12 mm, thickness 2 mm) for light reaction, setting a series of light time and taking
samples at specific time intervals to perform HPLC analysis ( **Text** S2 for lighting
equipment and specific steps). A blank control was performed during the experiment, and 3
parallel experiments were set up in each group.
*2.6 $^3OM^*$ and $^1O_2$ measurements and calculation of quantum yields*
In order to determine the production of $^3C^*$ and $^1O_2$, TMP (20 μM) and FFA (20 μM)
were used as capture agents for $^3C^*$ and $^1O_2$, respectively (Halladja et al., 2007; Dalrymple
et al., 2010). The two capture agent solutions have almost no loss under the direct irradiation
of the light source (**Fig.**S7). The probe concentration was analyzed by a high performance
liquid chromatography (LC-100, WuFeng, Shanghai) equipped with a C-18 column
(4.6×250 mm, 5 µm, Xuanmei). The acetonitrile (ACN) and ultrapure water were used as the
mobile phase, and the flow rate is 1 mL min$^{-1}$. The detection condition of TMP is mobile
phase ACN/Water = 50/50, and the detection wavelength is 210 nm; while for FFA, the
related terms are 30/70 and 219 nm, respectively.
The sorbic alcohol (1 mM) is used as a high-energy quencher to quench high-energy
$^3C^*$ (Zhou et al., 2017a), and combine the $\Phi_{1O2}$ to quantify the energy distribution of
different $^3C^*$. The reaction consumption rate of TMP ($k_{TMP}$) and the calculated quantum





yield coefficient ($f_{TMP}$) of the triplet state are used to reflect the $^3C^*$ generation rate and
ability. According to previous articles, the $\Phi_{1O2}$ and $f_{TMP}$ are calculated as follows
(Dalrymple et al., 2010; Bodhipaksha et al., 2015; Mostafa and Rosario-Ortiz, 2013).

$$\phi_{^1O_2} = \frac{R_{^1O_2}}{R_a} \tag{1}$$

$$f_{TMP} = \frac{k_{TMP}}{R_a} \tag{2}$$

$$R_a = \sum_\lambda \frac{I_\lambda(1-10^{-l\alpha_\lambda})}{l} \tag{3}$$

$$R_{^1O_2} = \frac{k_{FFA} \times k_d}{k'_{FFA}} \tag{4}$$

$R_{^1O_2}$ is the rate of generation of singlet oxygen (M s$^{-1}$); $R_a$ is the light absorption rate
(einsteins cm$^{-3}$ s$^{-1}$); $k_{TMP}$ and $k_{FFA}$ are the pseudo-first order reaction rate constant consumed
by TMP and FFA (s$^{-1}$); $I_\lambda$ is the photon flux at $\lambda$ wavelength (Einsteins cm$^{-2}$ s$^{-1}$ nm$^{-1}$), as
shown in S3 for specific calculation; $a_\lambda$ represents the unit absorbance of the solution at the
wavelength of $\lambda$ (cm$^{-1}$); $l$ is the thickness of the solution (0.1 cm); $k'_{FFA}$ is the rate constant of
$^1O_2$ and FFA reaction (1.2×10$^8$ M$^{-1}$ s$^{-1}$); $k_d$ is the rate constant of water quenching $^1O_2$
(2.4×10$^5$ s$^{-1}$). The wavelength range of light radiation for calculation in this study is from
320 nm to 600 nm.
*2.7 Reactive oxygen detection ($^1O_2$, ·OH)*
In order to explore the characteristics and mechanism of ROS generated by $^3C^*$, the
combination of free radical capture technology and electron paramagnetic resonance
spectrometer (EPR, MS5000, Freiberg) was used to detect two ROS including $^1O_2$ and ·OH.
The concentration of the experimental sample solution is controlled to 100 mg-C/L, using
TEMP as $^1O_2$ capture agent, DMPO as a capture agent for ·OH. The concentration of the
capture agent after mixing is 10 mM. When capturing $^1O_2$, the illumination time is 60 min,
and when capturing ·OH, the illumination time is 10 min. At the same time, the 0 point of
illumination is set upand the non-illumination control group is carried out. In this study,
Sorbic alcohol and histidine (both 50 mM after mixing) were used as quenchers of $^3C^*$ and
$^1O_2$, respectively, to explore the generation mechanism of $^1O_2$ and ·OH.



The EPR detection parameters for the $^1O_2$ and ·OH are the same: magnetic field
strength, 330-342 mT; detection time, 180 s; modulation amplitude, 0.2 mT; number of
detections, 1; and microwave intensity, 8.0 mW.

## 3. Results and Discussion


*3.1 Carbon composition of different polar aerosols*
Organic carbon is the main precursor for the formation of triplet states in aerosols, and
the composition of organic carbon is different between components of different polarities.
The results of carbon analysis are shown in **Fig.**1. The OC values of environmental $PM_{2.5}$
samples in winter is in the range of (19.51-44.97) µg-C/m$^3$, of which the WSOC component
accounts for about half of the OC, mainly the HP-WSOC component (25%-37%). The
average concentration of HULIS-C in this study is (3.39±1.38) µg·C/m$^3$, and the
concentration is between clean areas and areas severely affected by biomass burning
(Nguyen et al., 2014; Wang et al., 2017). Affected by source and environmental conditions,
the composition and concentration of HULIS vary greatly, generally accounting for 8-74%
of WSOC (Feczko et al., 2007). In this study, HULIS accounts for 26% of WSOC, which is
at a moderate level. The water-insoluble organic matter MSOC accounts for about 20% of
the OC of the $PM_{2.5}$ sample. This result indicates that there is a considerable part of the
water-insoluble organic carbon in the particulate sample. It is worth noting that the content
of water-insoluble organic carbon may be higher, considering that 34% of the organic carbon
has not been extracted (**Fig.**1 b1).
The MSOC in the primary combustion sample is generally higher than that in the actual
PM sample. In the biomass sample, 35%-64% of the components in OC are MSOC, while
the HP-WSOC and HULIS in OC are basically the same at 19%-22%. There are very few
water-soluble components in coal combustion samples. Nearly 90% of OC components are
MSOC. MSOC should be macromolecular organics, especially tar-like substances formed
during coal combustion. Whether it is coal combustion or biomass combustion, the content
of coking organic carbon in thermo-optical analysis is significantly higher than that of actual
atmospheric samples (**Fig.**S11).

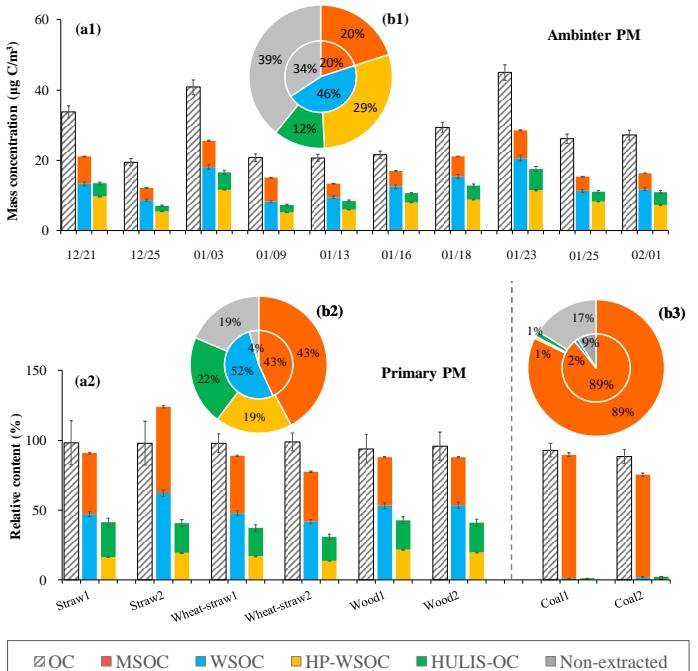

**Fig.1** The content of organic carbon in different polar components of atmospheric aerosol. (a1) and (a2) represent the organic carbon content in actual atmospheric aerosol samples and primary combustion source samples, respectively; (b1), (b2) and (b3) represent the average relative content of organic carbon of different polarity groups in actual atmospheric aerosols, biomass combustion aerosols, and coal combustion aerosol samples, respectively. The error bars represent the relative standard deviations obtained for a set of three parallel samples.

*3.2 Triplet generation ability of aerosols with different polarities*

The characteristics of triplet generation of components with different polar in aerosol are different. The pseudo-first order reaction kinetic rate constant $k_{TMP}$ of the triplet state and TMP is usually used to characterize the rate of formation of $^3C^*$, and the triplet quantum yield coefficient ($f_{TMP}$) reflects the ability of the triplet state to be generated (Ervens et al., 2011; Zhou et L., 2017b). **Fig.**2(A)(B) shows the average attenuation curve and average $k_{TMP}$ value of the reaction of different components with TMP under illumination conditions. The $k_{TMP}$ of each sample is shown in **Table** S5, and the average TMP attenuation fitting curve is shown in **Fig.**S16. The results show that the aerosol components with different polarities exhibit different generation rates of $^3C^*$. The $k_{TMP}$ of the components with different polar in the actual PM sample is between 0.004-0.017 min$^{-1}$, and the $k_{TMP}$ of the HP-WSM component is the lowest. In the biomass combustion samples, all polar components did not





significantly consume TMP; although HP-WSM in coal samples did not consume TMP,
HULIS and MSM components were consumed significantly, with an average $k_{TMP}$ value of
0.054 min$^{-1}$. **Table 1** lists the $f_{TMP}$ values of the components with different polar of each
sample. The $f_{TMP}$ value of the HULIS component of the atmospheric PM sample is between
(35-180) M$^{-1}$, which is higher than the HP-WSM component (5-90) M$^{-1}$ and MSM
component (14-70) M$^{-1}$, indicating that the HULIS has the strongest ability to generate
triplet states.

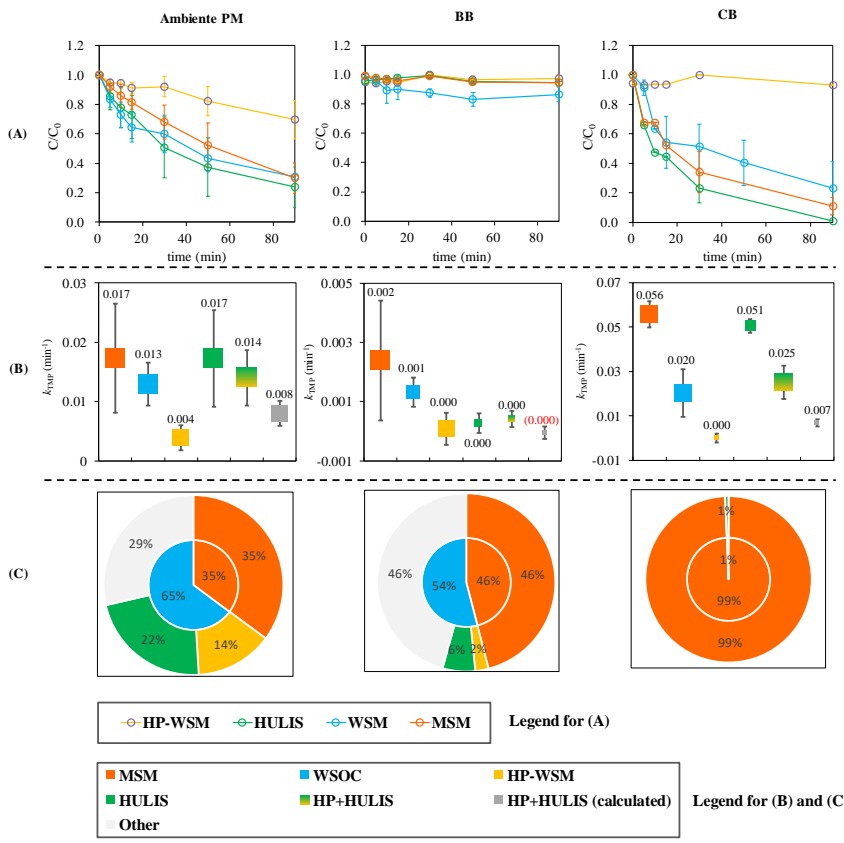


**Fig.2** (A) The average attenuation curve of TMP consumption by aerosol samples of different components; (B)
the triplet generation rate constant; (C) the average relative contribution to the overall triplet generation rate of
the aerosol. The error bar represents one standard deviation of the experimental values of all samples.

**Table 1.** Formation rate constant ($k_{TMP}$), quantum yield coefficient ($f_{TMP}$) and singlet oxygen quantum yield ($\Phi_{1O2}$)
of $^3C^*$ in the different polar components of each sample.





| Sample number | HP-WSM | | | HULIS | | | MSM | | |
|---|---|---|---|---|---|---|---|---|---|
| | $k_{TMP}$ | $f_{TMP}$ | $\Phi_{1O2}$ | $k_{TMP}$ | $f_{TMP}$ | $\Phi_{1O2}$ | $k_{TMP}$ | $f_{TMP}$ | $\Phi_{1O2}$ |
| | (min⁻¹) | (M⁻¹) | (%) | (min⁻¹) | (M⁻¹) | (%) | (min⁻¹) | (M⁻¹) | (%) |
| 2019/12/21 | 0.005 | 51.9 | 6.63 | 0.020 | 89.9 | 4.57 | 0.007 | 0.007 | 7.43 |
| 2019/12/25 | 0.006 | 56.9 | 5.78 | 0.025 | 158.3 | 7.60 | 0.014 | 0.014 | 11.91 |
| 2020/1/3 | 0.005 | 32.0 | 2.78 | 0.013 | 35.6 | 3.43 | 0.018 | 0.018 | 5.40 |
| 2020/1/9 | 0.006 | 71.5 | 9.23 | 0.021 | 82.7 | 6.45 | 0.033 | 0.033 | 14.37 |
| 2020/1/13 | 0.001 | 5.2 | 5.15 | 0.037 | 167.6 | 4.68 | 0.032 | 0.032 | 7.08 |
| 2020/1/16 | 0.002 | 24.5 | 6.53 | 0.025 | 140.9 | 6.82 | 0.022 | 0.022 | 8.44 |
| 2020/1/18 | 0.006 | 60.4 | 4.39 | 0.013 | 75.8 | 4.60 | 0.007 | 0.007 | 6.76 |
| 2020/1/23 | 0.008 | 91.8 | 11.92 | 0.010 | 43.7 | 3.57 | 0.015 | 0.015 | 7.03 |
| 2020/1/25 | 0.002 | 39.5 | 3.59 | 0.018 | 102.9 | 5.88 | 0.011 | 0.011 | 6.42 |
| 2020/2/1 | 0.002 | 22.6 | 6.79 | 0.009 | 40.3 | 5.37 | 0.028 | 0.028 | 8.29 |
| Wheat-straw1 | 0.000 | 0.00 | 3.31 | 0.000 | 4.6 | 1.70 | 0.000 | 0.000 | 4.89 |
| Wheat-straw2 | 0.000 | 0.00 | 2.18 | 0.000 | 5.4 | 1.49 | 0.000 | 0.000 | 6.34 |
| Rice-straw1 | 0.000 | 0.00 | 5.97 | 0.001 | 0.0 | 3.09 | 0.005 | 0.005 | 7.10 |
| Rice-straw2 | 0.000 | 0.00 | 3.66 | 0.001 | 0.0 | 4.66 | 0.004 | 0.004 | 4.75 |
| Wood1 | 0.001 | 16.3 | 2.45 | 0.000 | 2.0 | 1.95 | 0.001 | 0.001 | 8.80 |
| Wood2 | 0.001 | 10.3 | 2.40 | 0.000 | 1.5 | 3.02 | 0.003 | 0.003 | 7.61 |
| coal 1 | 0.000 | 3.2 | 1.49 | 0.047 | 270.8 | 38.85 | 0.050 | 0.050 | 25.92 |
| coal 2 | 0.000 | 0.0 | 1.47 | 0.054 | 417.2 | 34.18 | 0.062 | 0.062 | 48.77 |
| Ambient samples | 0.004 | 45.64 | 6.28 | 0.02 | 93.78 | 5.30 | 0.019 | 0.019 | 8.31 |
| (mean±SD) | ±0.002 | ±24.54 | ±2.56 | ±0.01 | ±45.97 | ±1.31 | ±0.009 | ±0.009 | ±2.61 |
| Primary samples | 0-0.001 | 0-16.3 | 1.5-6.0 | 0-0.05 | 0-417 | 1.5-38.9 | 0-0.06 | 0-0.06 | 4.8-48.8 |
| (range(mean)) | (0.0002) | (3.7) | (2.9) | (0.01) | (88) | (11.1) | (0.02) | (0.02) | (14.3) |


The ability to generate triplet states should be related to the chemical composition of
components with different polar. The HP-WSM component has a low ability to generate
triplet states, which may be related to the fact that the component contains more carboxylic
acids, alcohols and sugars and other small molecules with high polarity substances, which
cannot form triplet states (Bodhipaksha et al., 2015; Zhang et al., 2014). At the same time,
these small molecules are also easy to quench the triplet state. The combustion sample
showed different results from the actual sample (**Fig**.S13), although the particulate matter of
biomass combustion has a strong light absorption capacity (Lin et al., 2017). However, in
this study, the $f_{TMP}$ all the components with different polar in the biomass combustion
samples is small, and the $f_{TMP}$ of some components is even close to zero, indicating that the
light-absorbing organic matter of the biomass combustion source is not very capable of
generating ³C*. The HP-WSM component of the coal combustion sample has a very low



ability to generate $^3C^*$, while the HULIS and MSM components have higher $f_{TMP}$ values.
Especially the HULIS component, the content of the coal sample is very small, but it has the
highest generation ability of $^3C^*$ among all samples. From the results, the $k_{TMP}$ and $f_{TMP}$
trends of different polar components are basically the same. The difference is that the total
consumption rate of TMP $k_{TMP}$ for the MSM component is larger than that of the HULIS
component, but $f_{TMP}$ is smaller than that of HULIS, although MSM has stronger light
absorption capacity than HULIS (**Fig.**S13).

Colored organics and metal ions can change their original light-absorbing

characteristics through chelation (Wan et al., 2019; Kikuchi et al., 2017). Therefore, it is
expected that there will be an interaction between the HP-WSM and HULIS components
containing metal ions to affect the generation of triplet states. The difference between the
reaction $k_{TMP}$ value and the theoretical calculation value (the sum of the half of the respective
$k_{TMP}$ values) of the two components under the equal concentration and equal volume mixing
is comparatively studied. The result is shown in **Fig.**2(B). The mixed sample consumes more
TMP, and the actual environmental sample increases the generation rate of triplet by 1.8
times. This experiment demonstrates the existence of the interaction effect of photochemical
reactions between different components.

According to the proportion of different polar components in the sample OC, the

average contribution of different components to the total generation of $^3C^*$ is calculated, as
shown in **Fig.**2(C). The contribution of the WSM component to the total $k_{TMP}$ of PM is 65%.
After separating it into single components of HP-WSM and HULIS, they contribute 14%
and 22% of the triplet generation rate, respectively. Obviously, 29% has not been explained.
As the result described in the previous paragraph, this is partly due to the interaction
between HP-WSM and HULIS that promotes the generation of triplet states. The main
possible mechanism is that the metal elements in the system chelate with certain functional
groups of HULIS, forming more chromophores that can generate $^3C^*$, or reducing the
energy required for the original electronic transition, thereby increasing and enhancing the
generation rate of $^3C^*$. Note that this study cannot completely rule out the possibility that
inorganic salt components consume TMP, such as $\cdot SO_4^-$ (Fang et al., 2013). But the
experiments we added show that nitrate and sulfate do not consume TMP significantly under



light conditions. There is no relevant research on the formation of $^3C^*$ from water-insoluble
aerosol components. This study found that the contribution of MSM components to the
formation of $^3C^*$ can reach 35% in actual samples, and even more than 99% in coal-burning
samples, indicating that the contribution of water-insoluble organic carbon to aerosol triplet
photochemistry is significant.
*3.3 The energy distribution of triplet states of aerosols with different polarities*
Different organic matter in the aerosol can form the triplet state with different energy.
$^3C^*$ and $O_2$ in the environment can generate $^1O_2$ through energy transfer. Therefore, this
study uses the $\Phi_{1O2}$ to explore the energy distribution characteristics of $^3C^*$. The experiment
uses FFA as the quencher of $^1O_2$, combined with sorbic alcohol as the quencher of
high-energy $^3C^*$. After adding sorbic alcohol, the calculated $\Phi_{1O2}$ is considered to be the
contribution of low-energy $^3C^*$, which is different from the contribution of $\Phi_{1O2}$ when no
quencher is added. The difference is considered to be the contribution of high-energy $^3C^*$,
and the difference in reaction ability of $^3C^*$ with different energies is obtained (Zhou et al.,

2019).

**Fig.**3 shows the $^3C^*$ energy distribution of different polar components. According to the
results of $\Phi_{1O2}$, the high-energy triplet states in environmental samples accounted for
3.59%-11.92%, 3.57%-7.6%, 6.08%-14.37% in HP-WSM, HULIS and MSM components,
respectively. Among water-soluble components, high-energy $^3C^*$ accounts for an average of
close to 80%, which is significantly higher than that of natural water (33%) and sewage
(65%) samples (Zhou et al., 2019). This result is as stated in the introduction: atmospheric
samples and water environment samples are quite different in terms of source and
composition, and the conclusions of previous studies on the triplet state of water
environment samples are not necessarily suitable for the atmospheric PM samples. In MSM,
the proportion of low-energy $^3C^*$ increased, reaching an average of 52%, which should be
related to the organic matter containing polycyclic conjugated electrons in the
water-insoluble component. Especially for biomass combustion samples, the low-energy $^3C^*$
of different polar components has reached more than 50%. The $\Phi_{1O2}$ yield of the HP-WSM
component of coal samples is very low, while the $\Phi_{1O2}$ yield of the HULIS and MSM
components is the highest among all samples, reaching an average of 36% and 37%,
respectively. Unlikeother samples, most of the coal combustion samples are contributed by
high-energy $^3C*$.

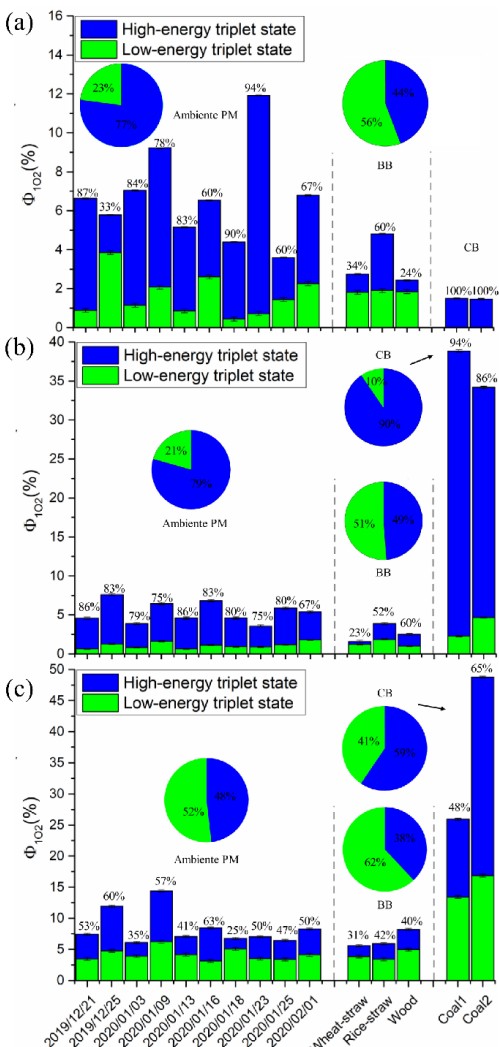


**Fig.3** $^3C*$ energy distribution in different polar sample components. (A) Sample HP-WSM component (b)

Sample HULIS component (c) Sample MSM component; pie chart represents the average proportion of high and

low energy triplet states; the percentage represents the $\Phi_{1O2}$ caused by the high energy triplet in the total $\Phi_{1O2}$.

The error bar represents one standard deviation of the experimental values of all samples.

High-energy $^3C*$ has higher energy and is more reactive than low-energy $^3C*$, and it is
easier to effectively collide with other substances at the molecular level. $^3C*$ can not only
generate $^1O_2$, but also react with other organic substances to generate active intermediates,



while the energy of low-energy $^3C^*$ only allows the conversion of $O_2$ to $^1O_2$. According to
previous reports, among the substances that can form triplet states, aromatic ketones and
other carbonyl compounds are likely to contribute a lot to high-energy $^3C^*$, while polycyclic
aromatic hydrocarbons and quinones are unlikely to be the main sources of high-energy $^3C^*$
(McNeill and Canonica, 2016; Kuznetsova and Kaliya, 1992).
*3.4. ROS generation caused by $^3C^*$*
$^3C^*$ can induce the generation of ROS, and the ability of triplet substances with
different polar components to generate ROS is different. This study uses EPR technology to
identify $^1O_2$ and $\cdot OH$ produced by different polar components (Blough and Zepp, 1995; Shi
et al., 2003; Cote et al., 2018). The triplet quencher is used to quantitatively judge the
contribution of the triplet process to the production of $^1O_2$. **Fig.**4 shows the EPR spectra of
the photochemical reaction products of different components of atmospheric aerosols. It can
be found that the output of $^1O_2$ is MSM>HULIS>WSM>HP-WSM, and HP-WSM is almost
not detected. When the triplet quencher sorbic alcohol is added, the signal of $^1O_2$ decreases
by 44%, 43% and 56%, respectively, indicating $^3C^*$ is an important precursor for the
photochemical generation of $^1O_2$. This study further explored the characteristics of $\cdot OH$
generated by water-soluble components under light conditions. The results showed that the
EPR spectrum of the HULIS sample showed a strong $\cdot OH$ signal, and the HP-WSM
component also showed a certain signal. When the triplet quencher is added, the $\cdot OH$ signal
is significantly reduced, and the signals of each component are reduced by 59% (WSM), 75%
(HULIS) and 26% (HP-WSM), respectively. This result demonstrates that $^3C^*$ is also an
important precursor of $\cdot OH$. Because the methanol in the MSM sample quenched the $\cdot OH$,
MSM was not tested.
The experimental results show that $^3C^*$ plays a leading role in the production of $^1O_2$
and $\cdot OH$. When there are triplet substances in the environment, the triplet state transfers
energy to $O_2$ molecules to form $^1O_2$, and $^1O_2$ may undergo complex reactions to
generate $\cdot OH$ (Chen et al., 2020; Mu et al., 2020). In this study, the genertation mechanism
of $\cdot OH$ varies with the components. For the HP-WSM component, it contains a lot of
inorganic substances such as metal ions, so it can form a Fenton-like system. For example,
in the presence of $H_2O_2$, $Fe^{2+}$, $Cu^{2+}$ will convert it into $\cdot OH$ (Shi et al., 2018). For the organic



component HULIS, the production of ·OH is mainly the effect of organic matter. HULIS
organic matter is excited to produce $^3C^*$ and then generates $^1O_2$, and finally, $^1O_2$ leads to the
formation of ·OH (**Fig.**S17). It should be noted that the possibility of triplet substances
directly producing ·OH cannot be ruled out. This potential mechanism is worth exploring in
the future.

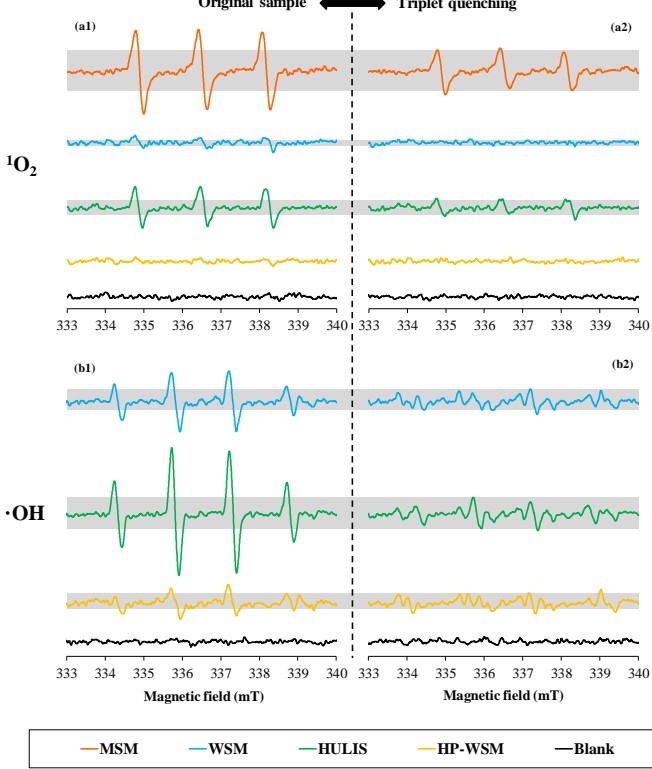


**Fig.4** Different aerosol components produce (a1) singlet oxygen and (b1) hydroxyl radicals under illumination
conditions, and the addition of triplet quencher (sorbic alcohol) produces (a2) singlet oxygen and (b2) hydroxyl
radicals feature.

The interaction between different components also affects the photochemical

generation of ROS. **Fig.**5 is the EPR signal spectrum of $^1O_2$ produced by mixing HP-WSM
and HULIS (All actual atmospheric samples mixed in equal amounts). The above results
have shown that the appearance of $^3C^*$ is promoted due to the interaction effect (**Fig.**2b).
Furthermore, the theoretical production of $^1O_2$ should be more than half of the sum of the
production of the two separate components, but **Fig.**5 shows a significant inhibitory effect



that the production of $^1O_2$ is greatly reduced after mixing. This is not the expected result, and
there may be a different mechanism than expected: the complex formed by metal ions and
HULIS increases the total rate of $^3C^*$ formation, but it is not sensitive to the reaction
pathway of energy transfer. It is easier to undergo chemical reactions through electron
transfer, resulting in a decrease in the production of $^1O_2$ (McNeill and Canonica, 2016; Wan
et al., 2019; Kikuchi et al., 2017; Blough and Zepp, 1995).

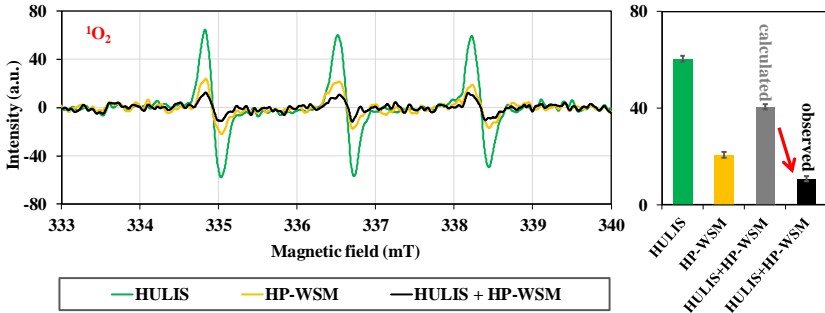


**Fig.5** The mixed sample of HP-WSM and HULIS produces the EPR spectrum of $^1O_2$. The Figure shows the
result of the actual atmospheric sample extract sample after 60 minutes of illumination (the signal of the control
sample placed in the dark for 60 minutes has been subtracted). Reaction conditions: the mixed sample is
illuminated for 60 minutes.
*3.5 The relationship between optical properties and $^3C^*$ generation*
The generation of $^3C^*$ should be related to optical characteristics. The study gives the
absorbance and fluorescence parameters of each sample (As shown in **Table** S5 and
**Fig.**S15). The results are similar to previous research reports. The non-polar components
have greater light absorption and fluorescence capabilities (Chen et al., 2017). Fluorescent
substances are the direct precursors for the formation of triplet states, which can be found by
the correlation between $k_{TMP}$ and standard fluorescence volume (NFV) (**Fig.**6). The sample
with stronger fluorescence intensity produces $^3C^*$ at a higher rate. However, the fluorescent
substances are still complex and diverse. In order to explore which chromophore is the most
important triplet precursor, this study further explored the structure-activity relationship
between chromophore types and triplet generation.




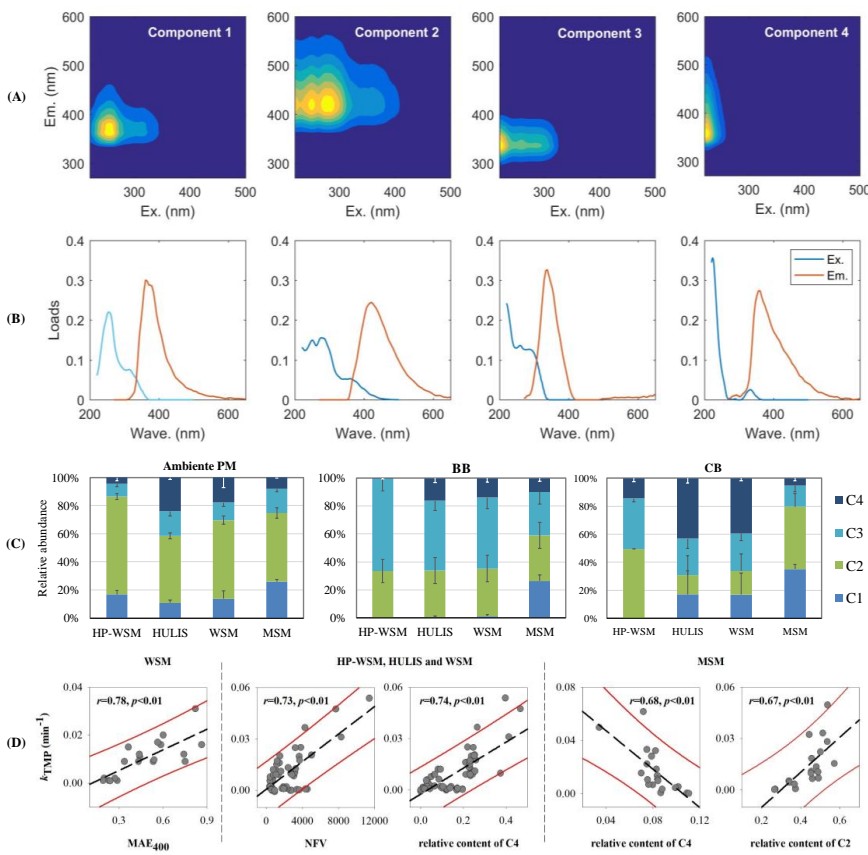

**Fig.6** (a and b) The four chromophores identified and (c) the relative content in different samples and

components, and (d) the correlation between $MAE_{400}$, NFV, C4, C2 and $k_{TMP}$. The error bar in the Figure (c)

represents one standard deviation of all sample values; the black dashed line in the Figure (d) represents the

linear fitting result, and the red line represents the 95% prediction interval.

This study identified four different chromophores C1-C4 from the samples using the

EEM-PARAFAC method (**Fig.**6). Refer to previous reports that the profile of C1 is similar

to low-oxidation HULIS; C2 may be highly oxidized HULIS; C3 may be Tryptophan-like

OM; and C4 is similar to Tyrosine-like chromophore (Chen et al., 2021; Murphy et al., 2013;

Chen et al., 2016; Chen et al., 2019). **Fig.**6(b) shows the proportions of the four

chromophores in each component sample. The results show that the HULIS chromophore is

the main light-absorbing organic matter in the water-soluble components, and compared

with the combustion source sample, the actual aerosol contains more highly oxidized HULIS,



which may be different from the actual atmospheric aerosol in the actual atmospheric
environment. That is probably related to the atmospheric aging process. Tryptophan-like OM
accounts for a relatively large proportion of combustion source samples. Recent studies have
shown that similar chromophores may be phenolic substances (Chen et al., 2020), and
biomass and coal combustion can emit a large number of phenolic substances.
The correlation between the different polar components $k_{TMP}$ and the composition of
chromophores were explored (**Fig.**6), and the correlation analysis between the quantum
yield $f_{TMP}$ and different types of chromophores was also carried out, and the results were
similar to $k_{TMP}$ Trend (**Fig.**S18). As shown in **Fig.**6, there is a positive correlation between
Tyrosine-like chromophores and $k_{TMP}$ in water-soluble components, and a negative
correlation in water-insoluble components. On the contrary, among the water-insoluble
components, the highly oxidized HULIS chromophore has an important role in promoting
the $^3C^*$ production of the MSM component. Analogous to the related literature, this
chromophore is likely to be a class of aromatic hydrocarbons containing oxygen functional
groups (Zhou et al., 2017; Zhou et al., 2019), structural substances are beneficial to the
formation of triplet states (**Fig.**S19). It is found that both high-energy $^3C^*$ and low-energy
$^3C^*$ in the MSM component have a positive correlation with the low-oxidation HULIS
chromophore, while the water-soluble component has no obvious correlation with the four
chromophores, which may be due to that the triplet reaction of the chromophore is different
(electron transfer and energy transfer). The low-oxidation HULIS chromophore may
generate $^3C^*$ through energy transfer, while the high-oxidation HULIS chromophore is
through electron transfer. For the MSM component, the triplet reaction is mainly an energy
transfer type reaction. HULIS chromophores play an important role. For example, quinones
may form low-energy $^3C^*$, while aromatic ketones may contribute a lot to high-energy $^3C^*$
(McNeill and Canonica, 2016; Kuznetsova and Kaliya, 1992).
**4. Environmental Implications**
$^3C^*$ plays an important role in the formation and aging process of atmospheric aerosols.
On the one hand, $^3C^*$ itself is reactive and can directly react with other substances. On the
other hand, it can produce $^1O_2$ and $\cdot OH$ and other ROS substances, which indirectly





participate in the generation reaction of aerosol components. In this study, the complex
aerosol samples were divided into three components including HP-WSM, HULIS and MSM
according to polarity, and their optical and photochemical reaction characteristics were
discussed respectively. This study demonstrated that the $^3C^*$ generation characteristics of
different polar components in atmospheric particulate matter samples are different. The
low-polar ity components have strong light absorption and fluorescence capabilities, and the
related $k_{TMP}$ is also enhanced. Among them, the $f_{TMP}$ of HULIS component is the strongest,
and the water-insoluble components contribute the most to the total $^3C^*$ generation. This
result means that the triplet photochemical reaction can enhance the heterogeneous aerosol
reaction. This study also demonstrated the complexity of the triplet types. The distribution of
different energy $^3C^*$ was indirectly investigated through $\Phi_{1O2}$, and it was found that
high-energy $^3C^*$ is the main form of $^3C^*$ (80%) in the water-soluble components of aerosols.
The high-energy $^3C^*$ in the insoluble components accounts for 50% or even lower, and the
high-energy $^3C^*$ plays a more important role in the production of $^1O_2$ and ·OH. Note that
there are only few reports on the energy distribution of the triplet state of aerosols so far.
Using EEM-PARAFAC technology, the structure-activity relationship between chromophore
composition and triplet generation was explored, and it was found that in the water-soluble
components, the chromophore previously defined as Tyrosine-like chromophore has a strong
correlation with the triplet generation rate. The highly oxidized HULIS chromophore in the
water-insoluble component plays an important role in promoting the production of $^3C^*$. This
result means that the photochemical reaction characteristics of different chromophores are
different, which ultimately determine the overall photochemical reactivity of the sample.
Finally, the obtained results also proved that there is a coupling effect of photochemical
reaction between HP-WSM and HULIS. After mixing, the production of $^3C^*$ is enhanced,
but the production of singlet oxygen is weakened. This result means that the photochemistry
of aerosol $^3C^*$ is not reacts alone, but is affected by the aerosol composition. In particular,
metal ions are most likely to undergo chelation with chromophore substances, thereby
changing the original optical and photochemical reaction characteristics of the chromophore.
It is necessary to explore this aspect in the future.



**Data availability.** All data that support the findings of this study are available in this article
and its Supplement or from the corresponding author on request.
**Supporting information.** Additional information as noted in the text, including three texts
(sampling information, photochemical experiment and calculation of quantum yields), six
tables and nineteen **Figures** (experimental detailed data).
**Author contributions.** DJ and QC designed the experiments and data analysis. DJ, JW and
LX performed sample collection. HL performed the EPR analysis. Other experiments are
performed by DJ, YQ, XF and TC polished the article. QC prepared the paper with the
contributions from all co-authors.
**Competing interests.** The authors declare that they have no conflict of interest.
**Acknowledgments.** We thank the National Natural Science Foundation of China (grant
number 41877354) and the Youth Science and Technology Nova Program of Shaanxi
Province (2021KJXX-36) for its financial support.

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
