# Peer review of "Supporting Information"

_Atmospheric Chemistry and Physics, 2021_

## Author Comment (AC1)

**Response to reviewers for the manuscript "Formation characteristics of aerosol triplet state and coupling effect between the separated components with different polarity" (acp-2021-842)**

We appreciate the comments from the editor and reviewer. According to the reviewer's comments,
we have revised this paper. The details are as follows. *The blue italics are comments of reviewer*. *The red italics are improvements and original text of manuscript.* The black font are responses.

**Response to Anonymous Referee #1**

10

35

Attention to detail is part of the scientific endeavour, and unfortunately this manuscript is missing important data to adequately assess the results. As it stands this manuscript is not fit for publication. Insufficient detail was provided, and other scientists cannot build upon the work, then the endeavour is fruitless. I encourage the authors to pay thorough attention to the details of their methods and results to clearly communicate to other scientists what was done and how to reproduce the data to subsequently build upon it. These steps include:

Build on existing literature. The introduction has little information on what has been done in this
 field so far. And key words and examples are missing, leaving the reviewer wondering why this research was done in the first place.

Many thanks to the reviewers for their suggestions. For the research progress in this field so far, the research method and formation process of triplet are introduced in the preface. The study of triplet states in complex systems starts from aqueous systems, so there are few references for aerosols that

- 20 can be used for reference. The novelty of this paper also makes up for the lack of related research in aerosol systems. If photochemical reactivity and different polar components can be linked, it will be helpful to further study photochemical reactions and control air pollution. The introduction part of the preface is more about the methods and ideas of studying triplet state. The photochemical properties of water-soluble components in aerosols have been published previously by our group.
- 25 This paper is an extension of previous research. Therefore, the description of the research background has been added as follow.

We have added "In previous studies, it was confirmed that water-soluble components in aerosols can generate triplet states, and is related to season and source." in introduction in the improved paper.

30 ≻ Chen, Q. C., Mu, Z., Xu, Li.: Triplet-state organic matter in atmospheric aerosols: Formation characteristics and potential effects on aerosol aging, *Atmos. Environ.*, 252, 118343, https://doi.org/10.1016/j.atmosenv.2021.118343, 2021.

We have deleted "The environmental organic matter components are composed of complex compounds with different polarities, which are expected to have different optical properties and photochemical reactivity." in introduction in the improved paper.

We have corrected "because the atmospheric chromophore is not a single substance, but a complex organic mixture, this also increases the difficulty of studying the generation mechanism of  $3C^*$  in

actual atmospheric environment." to "the environmental organic matter are complex, and we cannot study a certain substance alone, which may ignore the mutual influence that the system will have."

- 40 We have corrected "Due to the large differences in the sources and chemical processes of organic matter in the atmospheric environment and the water environment, the research conclusions on the 3C\* generation characteristics of water bodies may not be suitable for atmospheric aerosols." to "Different from the water environment, the source and chemical reaction of organic matter in aerosol is more complex, and the generation characteristics of 3C\* will be different."
- 45 2.Communicate the mechanisms correctly. Lines 39-40 incorrectly describe how triplet state organic matter forms from the intersystem crossing of singlet state organic matter. (The reader is left concerned after reading this section.)

We have corrected "After absorbing solar radiation, some organics will transition from the ground state to the excited singlet state ( $1C^*$ ), and then rapidly transition to  $3C^*$  through the intersystem."

50 to "The ground-states of some organic matters (C) can be excited to the singlet states (1C\*) under solar irradiation. And a part of 1C\* can transit rapidly into 3C\* by intersystem crossing (ISC)" in introduction in the improved paper.

3. Tell a story – why were PM and lab-generates filters made? What was the purpose of the comparison? Which hypotheses were being tested?

- (1) In winter, burning biomass and coal is the main heating method in northern China, especially in the rural areas. Coal burning is also the main source of energy in China. At the same time, previous studies have shown that the photochemical properties of aerosols from different sources are different. Therefore, we selected some lab-generates samples that are expected to share photochemical and optical properties with the PM samples.
- 60 (2) We have added "*because burning biomass and coal is the main heating method in northern China*" in section 2.2 in the improved paper.
  - Li, J., Chen, Q., Hua, X., et al.: Occurrence and sources of chromophoric organic carbon in fine particulate matter over Xi'an, China. *Science of the Total Environment*, 2020, 725, 138290, https://doi.org/10.1016/j.scitotenv.2020.138290.
- 4. Purify chemicals used, particularly the probes. (Line 107 explicitly states that all chemicals were used as is, and it is common practice in the community to distill furfuryl alcohol since it can easily dimerize and oxidize. Most papers in the field state that FFA is purified by distillation.)

Thanks to the reviewers for the questions and suggestions. As the reviewer said attention to detail is important part of the scientific endeavor, we also take into account the purity of the drug product

- 70 when planning the experiment. Section 2.1 of the text shows the purity and manufacturer information of FFA, and this purity sets the stage for a successful experiment. At the same time, there are literatures supporting that the chemical under this purity can be tested without purification. References are as follows:
- 75
- Bodhipaksha, L. C., Sharpless, C. M., Chin, Y. P.: Triplet photochemistry of effluent and natural organic matter in whole water and isolates from effluentreceiving rivers, *Environ. Sci. Technol.*, 2015, 49, 3453-3463, https://doi.org/10.1021/es505081w.

- Dalrymple, R. M., Carfagno, R. K., Sharpless, R. M.: Correlations between dissolved organic matter optical properties and quantum yields of singlet oxygen and hydrogen peroxide, *Environ. Sci. Technol.*, 2010, 44, 5824-5829, https://doi.org/10.1021/es101005u.
- 80
- Glover, C. M., Rosario-Ortiz, F. L.: Impact of halides on the photoproduction of reactive intermediates from organic matter, *Environ. Sci. Technol.*, 2013, 47, 13949-13956, https://doi.org/10.1021/es4026886.
- Mostafa, S., Rosario-Ortiz, F. L.: Singlet oxygen formation from wastewater organic matter, *Environ. Sci. Technol.*, 2013, 47, 8179-8186., https://doi.org/10.1021/es401814s.
- 85
- Vione, D., Minella, M., Maurino, V., et al.: Indirect photochemistry in sunlit surface waters: Photoinduced production of reactive transient species, *Chem. Eur. J.*, 2014, 20, 10590-10606, https://doi.org/10.1002/chem.201400413.

5. Describe all blanks in detail and show all the results. What were the field blanks, which controls were done? Line 158 is simply not good enough. Showing all the data of adequate background samples.

90 (1) Thanks to the reviewer for the suggestion. The experimental description here is not clear. "field blanks" means "background sample". We have elaborated the background sample preparation process in the improved paper.

(2) we have removed *Line 158* of original paper, and added "*Three background sample and three parallel samples were used for the experiments. The background sample is a sample that simulates*

- 95 the entire process from sampling to extraction with a blank membrane to deduct possible anthropogenic contamination. The parallel sample is to select one of the 10 aerosol PM2.5 samples, trim and extract three identical samples to verify the repeatability of the experiment. Each part of the experiment presents information about the background sample." in section 2.2 in the improved paper.
- 100 6. Use the most up-to-date rate constants. For example, FFA decay was revised back in 2017 by (Appiani et al., 2017) to be  $1.00'10^8 M^{-1} s^{-1}$ .

(1) We have performed calculations with new rate constants. After the change, it will slightly affect the original value and will not change the existing results and laws of this article.

(2) We have corrected the Table 1. in the improved paper.

|                 | HP-WSM               |                            |                      | HULIS                |                            |              | MSM                  |                            |              |
|-----------------|----------------------|----------------------------|----------------------|----------------------|----------------------------|--------------|----------------------|----------------------------|--------------|
| Sample number   | kTMP                 | frue                       | ${m \Phi}_{102}(\%)$ | krup                 | frum                       | Ø 102 | kno                  | frue                       | đ ion |
|                 | (min -1 ) | ( M -1 ) |                      | (min -1 ) | ( M -1 ) | €102
(%)  | (min -1 ) | ( M -1 ) | (%)          |
| 2010/12/21      | (                    | (1)1 )                     | 7.056                | (11111 )             | (111)                      | 5 404        | (11111 )             | 0.007                      | 0.016        |
| 2019/12/21      | 0.005                | 51.9                       | 7.956                | 0.020                | 89.9                       | 5.484        | 0.007                | 0.007                      | 8.916        |
| 2019/12/25      | 0.006                | 56.9                       | 6.936                | 0.025                | 158.3                      | 9.12         | 0.014                | 0.014                      | 14.292       |
| 2020/1/3        | 0.005                | 32.0                       | 3.336                | 0.013                | 35.6                       | 4.116        | 0.018                | 0.018                      | 6.48         |
| 2020/1/9        | 0.006                | 71.5                       | 11.076               | 0.021                | 82.7                       | 7.74         | 0.033                | 0.033                      | 17.244       |
| 2020/1/13       | 0.001                | 5.2                        | 6.18                 | 0.037                | 167.6                      | 5.616        | 0.032                | 0.032                      | 8.496        |
| 2020/1/16       | 0.002                | 24.5                       | 7.836                | 0.025                | 140.9                      | 8.184        | 0.022                | 0.022                      | 10.128       |
| 2020/1/18       | 0.006                | 60.4                       | 5.268                | 0.013                | 75.8                       | 5.52         | 0.007                | 0.007                      | 8.112        |
| 2020/1/23       | 0.008                | 91.8                       | 14.304               | 0.010                | 43.7                       | 4.284        | 0.015                | 0.015                      | 8.436        |
| 2020/1/25       | 0.002                | 39.5                       | 4.308                | 0.018                | 102.9                      | 7.056        | 0.011                | 0.011                      | 7.704        |
| 2020/2/1        | 0.002                | 22.6                       | 8.148                | 0.009                | 40.3                       | 6.444        | 0.028                | 0.028                      | 9.948        |
| Wheat-straw1    | 0.000                | 0.00                       | 3.972                | 0.000                | 4.6                        | 2.04         | 0.000                | 0.000                      | 5.868        |
| Wheat-straw2    | 0.000                | 0.00                       | 2.616                | 0.000                | 5.4                        | 1.788        | 0.000                | 0.000                      | 7.608        |
| Rice-straw1     | 0.000                | 0.00                       | 7.164                | 0.001                | 0.0                        | 3.708        | 0.005                | 0.005                      | 8.52         |
| Rice-straw2     | 0.000                | 0.00                       | 4.392                | 0.001                | 0.0                        | 5.592        | 0.004                | 0.004                      | 5.7          |
| Wood1           | 0.001                | 16.3                       | 2.94                 | 0.000                | 2.0                        | 2.34         | 0.001                | 0.001                      | 10.56        |
| Wood2           | 0.001                | 10.3                       | 2.88                 | 0.000                | 1.5                        | 3.624        | 0.003                | 0.003                      | 9.132        |
| Coal 1          | 0.000                | 3.2                        | 1.788                | 0.047                | 270.8                      | 46.62        | 0.050                | 0.050                      | 31.104       |
| Coal 2          | 0.000                | 0.0                        | 1.764                | 0.054                | 417.2                      | 41.016       | 0.062                | 0.062                      | 58.524       |
| Ambient samples | 0.004                | 45.64                      | 7.536                | 0.02                 | 93.78                      | 6.36         | 0.019                | 0.019                      | 9.972        |
| (mean±SD)       | ±0.002               | ±24.54                     | ±2.56                | ±0.01                | ±45.97                     | ±1.31        | ±0.009               | ±0.009                     | ±2.61        |
| Primary samples | 0-0.001              | 0-16.3                     | 1.8-7.2              | 0-0.05               | 0-417                      | 1.8-46.6     | 0-0.06               | 0-0.06                     | 5.7-58.5     |
| (range(mean))   | (0.0002)             | (3.7)                      | (3.48)               | (0.01)               | (88)                       | (13.32)      | (0.02)               | (0.02)                     | (17.16)      |

105 Table 1. Formation rate constant  $(k_{TMP})$ , quantum yield coefficient  $(f_{TMP})$  and singlet oxygen quantum yield  $(\Phi^I O_2)$  of  ${}^3C^*$  in the different polar components of each sample.

7. Caption all figures with thorough details. What do the colour wheels represent in Figure 1 and
2? The figure captions in the supporting information section contain so little data that figures are not understandable. (Striking examples include: Table S1 where all units are missing, copyright infringement for Figure S3, column with data not matching headings in Table S3 and in Table S4, poor resolution of Figure S10, no details in caption of Figure S12, no details of any of the 60 (!!) graphs in Figure S14.

115 Thanks to the reviewers for their suggestions. We have added relevant explanations or changes to the charts involved, which makes our articles better quality.

(1) The color wheel in Figure 1 represents the average relative content of organic carbon in different polar components, which is the percentage of the organic carbon content in the solution sample to the organic carbon content in the film sample.

120 (2) The color wheel in Figure 2 represents the proportion of  $k_{\text{TMP}}$  value of different polar components. We assume that the initial triplet reaction rate of the sample has only two parts, WSM and MSM. WSM is divided into HPWSM and HULIS. If the triplet state generation process between the components is independent, then the sum should be the same as that of WSM, otherwise there is a more complicated reaction mechanism.

125 (3) We have supplemented Table S1 with units.

We have modified Figure S3.

We have modified the heading of Table S3 to "Information of sample selection, trimming, and volume of post-extraction solutions".

We have modified the heading of Table S4 to "Sample concentrations used for absorbance and
fluorescence experiments of different polar components of the sample".

We have reprocessed the Figure S10.

135

145

150

We have added a detailed description of Figure S12, "The picture shows the results of carbon analysis of the samples using the OC/EC analyzer. Including four organic carbon components (OC1-OC4) and six elemental carbons (EC1-EC6), the red dotted line represents the optical pyrolyzed carbon produced during the analysis".

We have added a detailed description of Figure S14, "The picture shows the results of the fluorescence experiments using the samples with the concentrations listed in Table S4. The legend on the right shows the unit fluorescence intensity".

8. Rewrite the abstract to describe what was done: specific which aerosols were analysed and why,
and what methods were used to study the processes.

We have added "The experimental samples were selected from winter atmospheric  $PM_{2.5}$  samples. At the same time, due to the influence of the northern heating season in winter, and in order to highlight the generality of results, simulated combustion samples were also used in the experiment. Experiments using carbon analysis, probe method and electron paramagnetic resonance method to study the photochemical properties of different samples." in abstract in the improved paper.

**9. It is inaccurate to draw regressions through clusters of data as in Fig. 6.**

Thank you for your comment, but I don't understand what you mean of this comment. The purpose of our correlation analysis using the data in Figure 6 is to establish the relationship between the photochemical and optical properties of the samples. The results obtained so far can be interpreted with a certain degree of reliability.

10. Use acronyms consistent with the literature. (MSM and HP-SWM are not acronyms used by the community and are confusing, WSM in the community should be WSOC), use the word "probe" instead of capture agent.

- (1) Reviewers have extensive experience with research in this area. As the reviewer said, the abbreviations used in this article are slightly different from those in the references, because the WSM in this article refers to direct water solvent extraction without the process of desalination, which contains non-organic substances such as inorganic salts. This is also the key for this study to verify the coupling effect between inorganic salts and organic carbon. Therefore, the use of abbreviations such as WSOC may cause ambiguity.
- 160 (2) As suggested by the reviewer, we have replaced "*capture agent*" with "*probe*" in the full text.

11. Rewrite the environmental implications. As it stands that section reads as a summary. However, the implications section should extrapolate the findings to their impact on the environment and future work.

According to the comment, we have rewrote the environmental implications as following: "3C\*
plays an important role in the formation and aging process of atmospheric aerosols. On the one hand, 3C\* itself is reactive and can directly react with other substances. On the other hand, it can produce 1O2 and ·OH and other ROS substances, which indirectly participate in the generation reaction of aerosol components. This study demonstrated that the 3C\* generation characteristics of different polar components in atmospheric particulate matter samples are different. Low polar

- 170 components appear to play an important role in photochemical properties. Previous studies have suggested that water-soluble organic matter may play an important role in the generation of triplet states, but at present, water-insoluble substances or hydrophobic substances may contribute more to the generation of triplet states. This has certain implications for future research directions.
- The results of this paper show that the heterogeneous aerosol reaction can enhance the triplet photochemical reaction. The obtained results also proved that there is a coupling effect of photochemical reaction between HP-WSM and HULIS. What is the specific coupling effect between substances, and what is the coupling mechanism that is necessary to explore this aspect in the future."

180

185

12. The concentrations used by the authors of 100 mgC/L is very high. How did the authors obtain such high concentrations from aerosol filters (about a factor 10-100 more than typically found on PM filters in polluted environments)

The samples used in this paper are concentrated. After the extraction is obtained according to the normal extraction steps, the concentration at this time is not enough for triplet experiments, so the solution is evaporated by rotary evaporation and nitrogen blowing, and then quantitatively dissolved in a solvent to obtain the sample solution we used. The following figure shows the *concentration flow chart*. We have added this schematic to the supplementary material.

Fig.S4. Schematic diagram of solution sample concentration process.

13. How were the percentages in the first paragraph of the results calculated?

We have added "Relative content is determined by the ratio of the OC value in the sample solution
and the total OC value in the original film sample. The unit has been uniformly converted to the equivalent atmospheric mass concentration." in the description of Figure 1.

14. The authors must compare their results with the literature (ex. Line 213-214, "moderate level" compared to what?)

Since the content of HULIS fluctuates greatly according to different seasons and regions, it is impossible to generalize its content level, so this sentence in the improved paper has been deleted.

We have deleted "In this study, HULIS accounts for 26% of WSOC, which is at a moderate level."

15. The EPR results are promising, but how do these measurements compare with quantitative methods?

EPR is the most direct and effective technical means to detect the generation of free radicals. Probe 200 methods are somewhat non-specific and can lead to overestimate results. The EPR method can aid in the illustration. In this study, we did not use EPR to quantitatively study free radical production. We used it to directly verify free radical production and relative comparison.

**16. Show the blanks/controls in Fig. 5**

205

In order to remove the artificial interference during the experiment, the data curve results have been processed to subtract the value of the background sample. In Figure 5, the purpose is to verify whether there is a coupling effect between HPWSM and HULIS. The figure below is a comparison chart with the background signal. We have added the Figure S22 in the SI.

Fig.S22. EPR diagrams of different polar components and their interactions for produces 1O2.

**210 **Response to Anonymous Referee #2**

225

The authors examine the photogeneration of triplet excited states  $({}^{3}C^{*})$  in extracts of fine particles collected from the atmosphere and from several emissions sources (e.g., wood burning). The novel aspect of the work is examining triplet generation in three different polarity fractions of the PM: high-polar water soluble material (HP-WSM), water-soluble humic-like substances (HULIS), and

- 215 lower polarity methanol soluble material (MSM). They also measure the formation of singlet molecular oxygen ( ${}^{1}O_{2}*$ ) and hydroxyl radical (OH) in some of their samples. The broad idea of examining the photoreactivity of chromophores as a function of polarity is interesting. There are also some interesting results in the manuscript. But it's not clear that the results can be trusted since there are several important experimental errors, described below. The manuscript also
- 220 suffers from a frustrating lack of care in the writing, which makes the work very difficult to read and understand. Based on these important and widespread problems, I recommend that the manuscript be rejected.

We appreciate the comments from reviewer. We appreciate the positive evaluation of this work and very professional suggestions for improvement. According to the reviewer's comments, we have revised this paper. The details are as follows. *The blue italics are comments of reviews. The red italics are improvements and original text of reviews.* The black font are responses.

1. Quenching. The quenching experiments used to distinguish between high and low energy triplets (Section 3.3) are problematic. The authors examine the difference in 102\* formation with and without 1 mM sorbic alcohol (SA), which they believe will quench the high energy triplet states. But

- 230 this concentration of SA is not high enough to completely prevent high-energy triplet states from reacting with dissolved O2 to make 102\*. This leads to an incorrect assessment of high and low energy triplets. The authors need to use the kinetics of the  $3C^* + SA$  and  $3C^* + O2(aq)$  reactions, along with the SA and dissolved O2 concentrations, to understand what fraction of high-energy triplet states were actually quenched. They can then use this information to revise their estimate
- 235 of high- and low-energy triplets. Worse, the quenching experiments used to determine the contribution of triplets to OH generation (Section 3.4) are unusable. Sorbic alcohol, which they use as a triplet quencher, also reacts rapidly with OH to suppress its concentration. Thus examining the decline in OH with SA does not indicate the contribution of triplets to OH generation, as the authors believe, but shows the direct scavenging of OH. This is why the authors find an
- 240 unreasonably high contribution of triplet states to the generation of OH. The authors should calculate the expected decrease in OH signal based on the competition of OH between the EPR probe DMPO and SA. This likely explains most of the reduction in OH observed, indicating these results are unusable. Also in Section 3.4, the authors need to dig deeper into their 102\* results with the triplet quencher sorbic alcohol. Of course "...3C\* is an important precursor for the
- 245 photochemical generation of 102". This is well known. The more interesting question is why doesn't sorbic alcohol completely quench 102\* generation? The authors should examine the competition kinetics of 3C\* reacting with sorbic acid and 02 to see if their results make sense based on the concentrations of the two reactants.

(1) In this study, sorbic alcohol was selected as the quencher to quench the high-energy triplet state.
 Its concentration was obtained according to the reference (Zhou, H. X., et al., 2019). Secondly, we also used 2mM sorbic alcohol as a comparison, and verified that the concentration of 1mM can

completely quench high-energy triplet state, adding "1 mM, enough to quench" in the section 2.6 in the improved paper.

255

Zhou, H. X., Yan, S. W., Lian, L. S., Song, W. H.: Triplet-state photochemistry of dissolved organic matter: triplet-state energy distribution and surface electric charge conditions, Environ. Sci. Technol., 53, 2482-2490, http://dx.doi.org/10.1021/acs.est.8b06574, 2019.

(2) The picture below shows the same sample quenched by 1mM and 2mM SA, respectively (the MSM of the simulated coal-burning samples is selected because it has the strongest triplet ability).

Fig.S10. Quenching effect of different concentrations of quencher on samples.

(3) The reviewers' concerns are understandable and do explain the findings of this paper. Many previous studies used sorbic acid to quench the triplet state, and later studies reported that Sorbic alcohol is a quencher with stronger specificity and more stable properties than sorbic acid (Zhou, H. X., et al., 2017a), so this paper selects Sorbic alcohol as the quencher. And the ·OH signal is indeed weakened after quenching with sorbic acid in the previous article published by our research group (Chen, Q. C., et al., 2021). Therefore, we believe that the triplet state contributes to the generation

270

265

(4) The generation and quenching of triplet states is a dynamic process. They reach a dynamic equilibrium at a certain stage of the reaction, and the appearance is that the triplet is no longer generated and is reacted away. When  $O_2$  intervenes, a new balance will be reached, so each will not disappear. This is the amount of dynamic balance.

of hydroxyl groups, and it remains to be verified how much the contribution is.

- Zhou, H. X., Yan, S. W., Ma, J. Z., Lian, L. S., Song, W. H.: Development of novel chemical probes for examining triplet natural organic matter under solar illumination, *Environ. Sci. Technol.*, 51, http://dx.doi.org/10.1021/acs.est.7b02828, 11066-11074, 2017a.
- 275 ➤ Chen, Q. C., Mu, Z., Xu, Li.: Triplet-state organic matter in atmospheric aerosols: Formation characteristics and potential effects on aerosol aging, *Atmos. Environ.*, 252, 118343, https://doi.org/10.1016/j.atmosenv.2021.118343, 2021.

 2. TMP as a Measure of Triplets. The authors have ignored the potential inhibition of TMP decay by sample constituents, especially phenols and dissolved copper (Canonica, Photochem. Photobiol. Sci. 2008; Pan, ES&T, 2018). This inhibition slows the apparent decay of TMP, leading to an underestimate of triplet concentrations (or, here, k(TMP) and f(TMP)). Since both phenols and copper will be most enriched in the HP-WSM fraction, the slow TMP loss seen in this fraction could be a result of inhibition (which is an artifact) and not the result of low triplet concentrations. The authors should measure inhibition factors of TMP loss in different fractions. This inhibition of TMP

- 285 loss is a more likely explanation of their HP-WSM results than is the theory presented on page 12. TMP inhibition is probably also a major contributor to the apparent lack of TMP loss in the BB samples (Fig. 2). Previous work has shown that biomass burning PM generate high levels of triplets (e.g., Kaur, ACP, 2019), including the authors' own recent work showing that BB emissions had the highest triplet reactivity (Chen, Atm Env, 2021). I suspect that part/most of this "missing reactivity"
- 290 is due to inhibition of the TMP probe, which has been seen previously at DOC concentrations of 20 mg-C/L. Essentially, the oxidized TMP probe is reduced back to TMP by phenols in the BB extracts, reducing the apparent loss of TMP. See work by Canonica and others for a description of the inhibition. If this is the reason for the Fig. 2A BB results (which seems likely), it means that they're incorrect and that triplets are generated. Also, the authors appear to believe that oxidant probes
- 295 are perfectly specific, i.e., only react with the oxidant of interest, but this is not true. For example, TMP reacts with OH and 102\* as well as triplets. These interferences have often been minor in past studies, but this is not always the case. The authors should quantify OH and 102\* in some samples to see if they make significant contributions to TMP loss. Finally, for triplets the authors report k(TMP), the pseudo-first-order rate constant for TMP loss, and f(TMP), the quantum yield
- 300 coefficient for 3C\*. While these quantities are useful for comparing with past work, they're limited in value otherwise. It would be much better to estimate and report 3C\* production rates and quantum yields, which are more useful. The authors should think about how they could estimate these quantities based on their data.
- (1) Thanks to the reviewers for their valuable comments. It is undeniable that there are indeed some
   metal ions (eg, dissolved copper) that inhibit triplet formation. However, As mentioned in the article of Pan et al., when the copper ion reaches 50µM, it will have an inhibitory effect on the triplet state. But the content of copper in the actual aerosol is very low, even if the concentration is increased several times in this experiment, the content of dissolved copper in the sample is still at the nM level. So, this content of copper ion is not enough to have a fundamental impact on the experimental results.
- 310 We have added "It is worth noting that although HP-WSM contains a lot of inorganic salts, its concentration is still very low and will not have a negative effect on the experiment." in section 3.2.
- (2) It cannot be denied that the low triplet activity exhibited by the biomass samples in this paper is inconsistent with previous studies. It is even inconsistent with the results of previous articles of this
  research group. Through a comprehensive comparison from sample preparation to measurement process, it is found that the problem may be generated in the preparation of the combustion samples. About 2 g of raw material was weighed in the preparation of the simulated combustion samples in this study, which was several times higher in previous studies. Since there are more raw materials, there are more combustion products, which may cause more molecular species to polymerize to
- 320 form macromolecular species during the collection process. As a result, the physicochemical properties of the sample itself have been changed.

We have add "Weigh a certain mass of raw material samples for combustion (about 2 g), and collect spawns through a self-built combustion-gathering device." in the section 2.2.

- (3) The probe method to study triplet states is a simple and widely used method. And TMP is the most commonly used triplet probe. As the reviewer said, although it cannot be said that TMP is completely specific to the triplet state, it is generally recognized that it has strong stability and anti-interference. The references cited in this paper all use TMP as a triplet probe. At the same time, except for triplet, the content of other oxidizing substances (hydroxyl, singlet oxygen) contained in the sample is very small, which has little effect on the loss of TMP.
- 330 ➤ Bodhipaksha, L. C., Sharpless, C. M., Chin, Y. P.: Triplet photochemistry of effluent and natural organic matter in whole water and isolates from effluentreceiving rivers, *Environ. Sci. Technol.*, 49, 3453-3463, https://doi.org/10.1021/es505081w, 2015.
  - Halladja, S., Ter-Halle, A., Aguer, J. P.: Inhibition of humic substances mediated photooxygenation of furfuryl alcohol by 2,4,6-trimethylphenol: Evidence for reactivity of the phenol with humic triplet excited states, *Environ. Sci. Technol.*, 41, 6066-6073, https://doi.org/10.1021/es070656t, 2007.
  - Kaur, R., Anastasio, C.: First measurements of organic triplet excited states in atmospheric waters. *Environ, Sci. Technol.*, 52, 5218-5226, https://doi.org/10.1021/acs.est.7b06699, 2018.

(4) Very much agree with the reviewer's comments. The quantum yield is the most direct and effective method to evaluate the triplet generation ability. However,  $f_{TMP}$  is also widely used as an

evaluation index for triplet generation, which is also scientific and objective. Compared to quantum yield,  $f_{TMP}$  requires less experimental measurement.

3. The Writing. The manuscript is often difficult to read, sometimes because there are multiple ideas strung together without a logical flow or transitions, and sometimes because the text doesn't make sense. Examples of the latter include:

345 *line 80, "The cause of the formation of 3C\* is directly related to the chromophore."*

335

350

We have corrected "The cause of the formation of  $3C^*$  is directly related to the chromophore." to "Since the chromophore is the direct reason of triplet generation." in the improved paper.

lines 170-172, "The sorbic alcohol (1 mM) is used as a high-energy quencher to quench high-energy  $3C^*$  (Zhou et al., 2017a), and combine the  $\Phi(102)$  to quantify the energy distribution of different  $3C^*$ ."

We have corrected "The sorbic alcohol (1 mM) is used as a high-energy quencher to quench highenergy  ${}^{3}C^{*}$  (Zhou et al., 2017a), and combine the  $\Phi_{102}$  to quantify the energy distribution of different  ${}^{3}C^{*}$ ." to "Dissolved oxygen can be converted into singlet oxygen by energy transfer due to triplet state. Therefore,  $\Phi_{102}$  can be used to evaluate the energy distribution of the triplet state. The sorbic

355 alcohol (1 mM) is used as a quencher to quench high-energy  ${}^{3}C^{*}$  (Zhou et al., 2017a)." in the improved paper.

line 371: "For the organic component HULIS, the production of  $\cdot OH$  is mainly the effect of organic matter."

We have corrected "For the organic component HULIS, the production of •OH is mainly the effect
of organic matter." to "For HULIS components, the generation of •OH are only possible from organics." in the improved paper.

In addition, portions of the text are incorrect. For example, lines 122-123: "Note that the MSM [methanol-soluble material] here does not actually contain water-soluble substances, thus it represents water-insoluble organic matter." This is almost certainly not true: a significant portion

365 of the compounds in the particles will be soluble in both water and methanol. Another example is lines 264-265: "At the same time, these small molecules are also easy to quench the triplet state." The evidence I have seen – that small molecules do not quench triplet states – contradicts the authors' claim.

(1) The MSM here is obtained by water solvent extraction followed by methanol solvent extraction.
 At this time, most of the extracted substances are not easily soluble in water. In order to study from the perspective of polarity, they are called water-insoluble substances here.

(2) The expression "*lines 264-265*" means that energy can be released through physical processes without chemical reactions. The figure below shows a possible reaction pathway for the triplet state stated in the study by *Rosario-Ortiz* et al., which verify  $3C^*$  to lose energy through physical processes.

**375 processes.**

380

We have corrected "At the same time, these small molecules are also easy to quench the triplet state." to "3C\* release energy by physical means" in the improved paper.